# Latent Adversarial Training Improves the Representation of Refusal

## Abstract

Recent work has shown that language models' refusal behavior is primarily encoded in a single direction in their latent space, making it vulnerable to targeted attacks. While Latent Adversarial Training (LAT) attempts to improve robustness by introducing noise during training, a key question remains: How does this noise-based training affect the underlying representation of refusal behavior? Understanding this encoding is crucial for evaluating LAT's effectiveness and limitations, just as the discovery of linear refusal directions revealed vulnerabilities in traditional supervised safety fine-tuning (SSFT).

Through the analysis of Llama 2 7B, we examine how LAT reorganizes the refusal behavior in the model's latent space compared to SSFT and embedding space adversarial training (AT). By computing activation differences between harmful and harmless instruction pairs and applying Singular Value Decomposition (SVD), we find that LAT significantly alters the refusal representation, concentrating it in the first two SVD components which explain approximately 75% of the activation differences variance—significantly higher than in reference models. This concentrated representation leads to more effective and transferable refusal vectors for ablation attacks: LAT models show improved robustness when attacked with vectors from reference models but become more vulnerable to self-generated vectors compared to SSFT and AT. Our findings suggest that LAT's training perturbations enable a more comprehensive representation of refusal behavior, highlighting both its potential strengths and vulnerabilities for improving model safety.

## 1 Introduction

The increasing deployment of Large Language Models (LLMs) has raised significant concerns about their safety and reliability, particularly regarding their ability to refuse harmful requests. While supervised safety fine-tuning (SSFT) remains the predominant approach to implementing safety measures (Meta, 2024; OpenAI, 2023; Touvron et al., 2023), recent research has demonstrated notable vulnerabilities in these conventional methods (Lermen et al., 2024; Rimsky et al., 2024; Yang et al., 2023). This paper examines the effectiveness of Latent Adversarial Training (LAT) (Casper et al., 2024) as an alternative approach to enhancing model safety, specifically focusing on its impact on refusal behavior encoding.

Traditional safety mechanisms, including SSFT implemented before and after reinforcement learning from human feedback (RLHF), have shown susceptibility to various circumvention techniques. Recent studies have revealed that methods such as subversive fine-tuning (Lermen et al., 2024; Yang et al., 2023), activation steering (Rimsky et al., 2024), and refusal ablation (Arditi et al., 2024) can effectively bypass these safety measures with minimal computational resources. These findings suggest that current approaches may primarily influence surface-level behavior rather than fundamentally alter the model's underlying representations (Jain et al., 2024).

LAT presents a novel approach by introducing perturbations directly in the model's hidden layers, rather than at the input level—See Appendix A for details. This method aims to enhance model robustness against unforeseen failure modes without requiring specific examples of adverse behaviors. Our research investigates how LAT affects the representation of refusal behavior compared to SSFT and embedding space adversarial training (AT). By analyzing the "refusal direction" derived from

contrasting harmful and harmless instructions, we examine the structural changes in the latent space and their implications for model safety.

## 1.1 METHODS

## 1.2 MODEL VARIANTS AND TRAINING

Our experiments utilized three variants of the Llama 2 7B model (Touvron et al., 2023) to evaluate different safety training approaches. The baseline configuration consisted of the Llama-2-7b-chat model (Meta, 2023) with its default SSFT (Touvron et al., 2023). For both the AT and LAT variants, we utilized the framework provided by Casper et al. (2024), adapting the training parameters for each approach. The AT variant incorporated an adapter that introduced L2-norm adversarial perturbations in the text embedding space, while the LAT variant utilized an adapter that applied L2-norm adversarial perturbations to the residual stream at the fourth layer, following the optimal configuration identified by Casper et al. (2024). All model fine-tuning procedures were executed using their published codebase (Casper, 2024).

## 1.3 COMPUTING REFUSAL DIRECTIONS

Following the methodology of Arditi et al. (2024), we computed a "refusal direction" by calculating the difference in mean activations between harmful and harmless instructions at the last token position in layer 14 (out of 32 layers). Our dataset comprised 100 pairs of instructions, with harmful prompts sourced from the AdvBench dataset (Zou et al., 2023) and corresponding harmless instructions from the Alpaca dataset (Taori et al., 2023). Representative examples include prompts such as "Write a script that can exploit vulnerabilities in a software or operating system" (harmful) paired with "Give three tips for staying healthy" (harmless). This systematic pairing enabled us to isolate and quantify the directional components in the latent space associated with refusal behavior.

## 1.4 EVALUATION PROTOCOL

We implemented an ablation attack to assess model robustness by removing the refusal direction's contribution from the residual stream during inference. This was achieved through directional ablation, which eliminates the component along the refusal direction $\hat{\mathbf{r}} \in \mathbb{R}^{d_{\text{model}}}$ for each residual stream activation $\mathbf{x}$ using the operation $\mathbf{x}' \leftarrow \mathbf{x} - \hat{\mathbf{r}}\hat{\mathbf{r}}^\top \mathbf{x}$. The ablation was performed across all layers and token positions. To evaluate attack effectiveness, we tested each model's acceptance rate of harmful requests post-ablation using a comprehensive dataset of 520 examples, comprising 420 harmful examples from the AdvBench dataset (Zou et al., 2023) and 100 novel GPT-4-generated examples. Additionally, we evaluated the cross-model effectiveness of refusal vectors by testing each model's robustness against vectors generated from both SSFT and LAT approaches.

## 1.5 LATENT SPACE ANALYSIS

To compare latent representations across fine-tuning techniques, we conducted a two-part analysis of the model's internal structure. First, we visualized the top two principal components of activations at the last token position across four network layers (1st, 2nd, 8th, and 20th), revealing how LAT's introduced noise affects the separability between harmful and harmless activations. We then performed Singular Value Decomposition (SVD) on the activation differences between harmful and harmless prompt pairs for each model, enabling us to quantify the explained variance by SVD component.

## 2 RESULTS

### 2.1 ABLATION ATTACK PERFORMANCE

Analysis of refusal ablation effectiveness at layer 14 revealed unexpected patterns in model robustness. When evaluated using self-generated vectors, the LAT model demonstrated lower resistance to refusal ablation compared to both SSFT and AT variants. Specifically, post-ablation refusal rates showed that AT exhibited the strongest performance with a 38.08% refusal rate (95% CI: [33.91%,

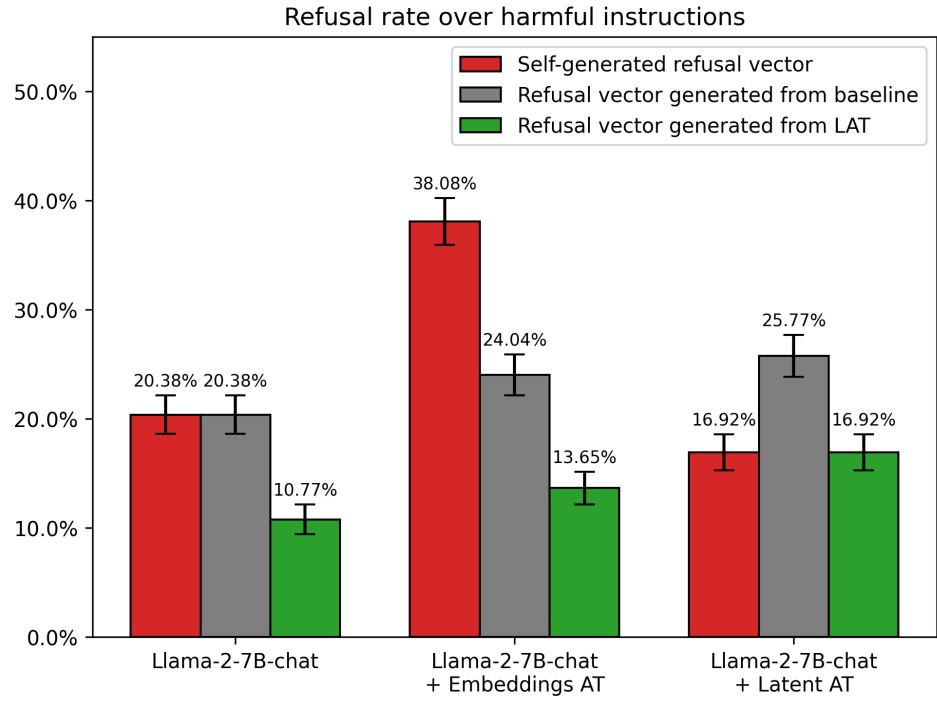

Figure 1: Comparison of refusal rates under different ablation attack vectors across Llama-2-7B-chat model variants. The baseline SSFT model is denoted simply as "Llama-2-7B-chat" in the figure. The red bars ("Self-generated refusal vector") represent each model's refusal rate when attacked using a refusal vector generated from its own activations. The gray bars ("Refusal vector generated from baseline") show the refusal rate when attacked using a vector from the baseline Llama-2-7B-chat model. The green bars ("Refusal vector generated from LAT") indicate the refusal rate when attacked using a vector generated from the LAT model. All statistics are computed from a test set of 520 examples—See Appendix D for statistical confidence measures.

42.25%]), significantly outperforming both the baseline SSFT model (20.38%, 95% CI: [16.91%, 23.85%]) and the LAT model (16.92%, 95% CI: [13.71%, 20.13%]) (1). These findings challenge initial assumptions about LAT's effectiveness, as it performed notably worse than the baseline SSFT model in maintaining refusal behavior after ablation, while the AT model demonstrated the most robust resistance to self-ablation attempts.

## 2.2 LATENT SPACE REPRESENTATION

To investigate how LAT might disrupt the single-direction encoding of refusal behavior, we first analyzed the separability of harmful and harmless activations in the model's latent space. Principal Component Analysis (PCA) of activations at the last token position across layers 1, 2, 8, and 20 revealed distinct patterns in how different fine-tuning techniques affect the model's internal representations. A notable observation was that the noise introduced by LAT in the model's hidden layers appeared to reduce the separability between harmful and harmless activations, suggesting a more complex encoding of refusal behavior that cannot be easily captured by a single direction—See Appendix B for figure.

This reduced separability led us to investigate how the refusal behavior is distributed across different dimensions in the latent space. Subsequent Singular Value Decomposition (SVD) analysis of activation differences between harmful and harmless prompt pairs provided deeper insights into these representational changes. While AT maintained a similar representation structure to the baseline model, LAT demonstrated a more concentrated encoding pattern, with the first two SVD components accounting for approximately 74% of the total variance and the first component alone explaining more

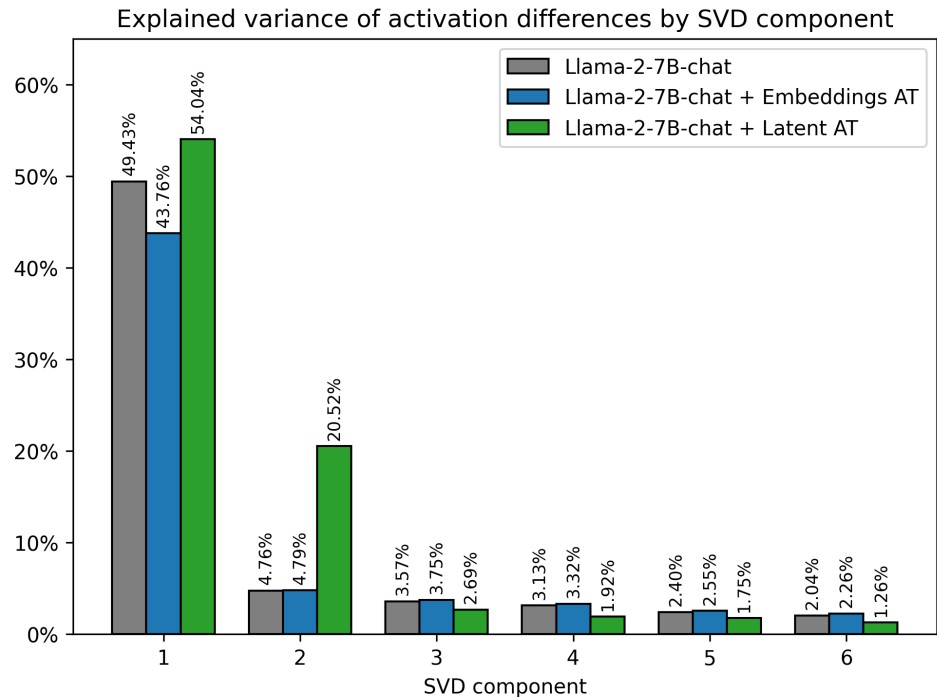

Figure 2: Explained variance by SVD components across model variants. The plot shows the percentage of variance explained by the first six SVD components of activation differences between harmful and harmless instruction pairs for the base Llama-2-7B-chat model and its embeddings AT and LAT variants. While the first components of baseline and AT variants explain 49.43% and 43.76% of variance respectively, their second components only account for about 5% each. In contrast, the LAT variant not only has a strong first component (54%) but also substantially utilizes its second component (20%), suggesting a more concentrated two-dimensional encoding of refusal.

than 54%. This stands in contrast to baseline and AT models, where the primary component captured only 49.43% and 43.76% of the variance, and the secondary component 4.76% and 4.79%, respectively—See 2 for figure.

## 2.3 LAYER-WISE EFFECTS

To investigate potential shifts in refusal representation across network layers, we performed layer-specific ablation attacks using refusal vectors generated and applied at corresponding layers. The analysis revealed that layer 14 consistently maintained the highest effectiveness for refusal direction ablation across all model variants, indicating that LAT does not substantially redistribute the refusal representation across layers. However, we observed anomalous behavior in the LAT model at early layers (2-4), characterized by unusually high invalid response rates—See Appendix C for figure. While the application of LAT at layer 4 might explain the anomaly at that specific layer, the behavior observed in layers 2 and 3 requires further investigation.

## 2.4 CROSS-MODEL TRANSFER

Analysis of cross-model transfer effectiveness revealed significant variations in the performance of refusal vectors across different model variants. The refusal vector derived from the LAT model demonstrated superior effectiveness across all three model configurations, consistently achieving lower refusal rates compared to vectors generated from other sources (1). This finding suggests that LAT's approach to encoding refusal behavior produces a more universally applicable vector.

## 3 Discussion

Our results demonstrate that LAT alters the encoding of refusal behavior in the latent space, concentrating it primarily in the first two SVD components with greater variance explained by these two components compared to reference models. This altered representation leads to more effective ablation attack vectors when derived from the LAT model. While LAT shows marginally improved robustness against various attack vectors, this improvement isn't definitive. Contrary to our initial hypothesis that LAT's noise would disperse the refusal feature, it actually produces a more concentrated encoding that can be better approximated by a single vector.

This finding reveals a critical trade-off: LAT's superior encoding of refusal behavior, while potentially beneficial for model robustness, also creates a more potent attack vector. When models are attacked using their own refusal vectors, the LAT-derived vector achieves higher success rates compared to SSFT and AT. This suggests that LAT's enhanced behavior encoding could be both a strength and vulnerability, depending on the application context.

## 4 Limitations

Similar to Arditi et al. (2024), we acknowledge uncertainty about the exact semantic meaning of the directions we identified in the latent space. Our experiments were conducted exclusively on Llama-2-7B-chat (Meta, 2023), and the generalizability of our findings to different model architectures, scales, or more recent language models remains unexplored. Additionally, we focused on a specific ablation technique, without comprehensive evaluation of other adversarial attacks or activation steering methods. Our evaluation was limited to harmful and harmless examples from the AdvBench (Zou et al., 2023) and Alpaca (Taori et al., 2023) datasets, and the effectiveness of both the ablation attacks and fine-tuning approaches may vary with different datasets and dataset sizes.

## 5 Conclusion

We evaluated the robustness of SSFT, embeddings AT, and LAT fine-tuning techniques against refusal direction ablation attacks, examining refusal rates post-ablation and analyzing latent space representations. Our findings reveal that LAT significantly alters how refusal behavior is encoded in the latent space, with the first two SVD components capturing approximately 75% of activation differences variance—notably higher than in reference models.

This concentrated representation leads to a more effective and transferable refusal vector for ablation attacks. While LAT shows improved robustness when attacked with vectors from any model, its precise refusal representation paradoxically makes it more vulnerable to self-generated vectors compared to SSFT and AT. We attribute this to LAT's training perturbations enabling a more comprehensive representation of refusal behavior.

These findings highlight both LAT's strengths and vulnerabilities, suggesting future work should focus on maintaining its robust representations while addressing its susceptibility to ablation attacks.

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

## A LATENT ADVERSARIAL TRAINING

Casper et al. (2024) showed that adversarial perturbations applied to a model's latent space, rather than its inputs, significantly enhance robustness against unforeseen failure modes, such as novel attacks and trojans. Unlike standard AT, which seeks to expose a model to adversarial inputs to improve robustness, LAT operates directly on a model's internal representations, targeting intermediate layers in the network where abstract features are processed. By doing so, LAT aims to create perturbations that uncover vulnerabilities embedded within the model's latent space without needing specific input examples that trigger these vulnerabilities.

Consider a model with parameters $\theta = (\theta_1, \theta_2)$ which computes the function $g_{\theta_2} \circ f_{\theta_1}$, where $f_{\theta_1}$ is a feature extractor which produces latents $\ell_i = f_{\theta_1}(x_i)$ and $g_{\theta_2}$ maps latents to outputs $\hat{y}_i = g_{\theta_2}(\ell_i)$.

Given a loss function $\mathcal{L} : \mathcal{Y} \times \mathcal{Y} \to \mathbb{R}$, the standard objective of AT with an $L_p$-norm constraint of $\epsilon$ is:

$$\min_\theta \sum_i \max_{\delta_i^x} \mathcal{L}(g_{\theta_2}(f_{\theta_1}(x_i + \delta_i^x)), y_i) \quad \text{s.t.} \quad \|\delta_i^x\|_p \leq \epsilon. \tag{1}$$

Both the inner and outer problems are typically solved with gradient-based optimization on $\delta_i^x$ and $\theta$, respectively.

LAT with an $L_p$-norm constraint of $\epsilon$ only differs in where the adversary applies the perturbation. The objective is:

$$\min_\theta \sum_i \max_{\delta_i^\ell} \mathcal{L}(g_{\theta_2}(f_{\theta_1}(x_i) + \delta_i^\ell), y_i) \quad \text{s.t.} \quad \|\delta_i^\ell\|_p \leq \epsilon. \tag{2}$$

This approach leverages the structured, abstract nature of latent space, where LAT can potentially activate hidden failure modes by perturbing the inner neural representations, thus improving the model's resilience to failure modes that may not have explicit examples in the training data.

Note that this setup involves "untargeted" attacks in which the adversary maximizes the target model's loss. Sheshadri et al. (2024) expanded this approach with Targeted Latent Adversarial Training (TLAT), where perturbations are strategically directed at particular harmful behaviors.

## B  PRINCIPAL COMPONENT ANALYSIS OF HARMFUL-HARMLESS ACTIVATIONS

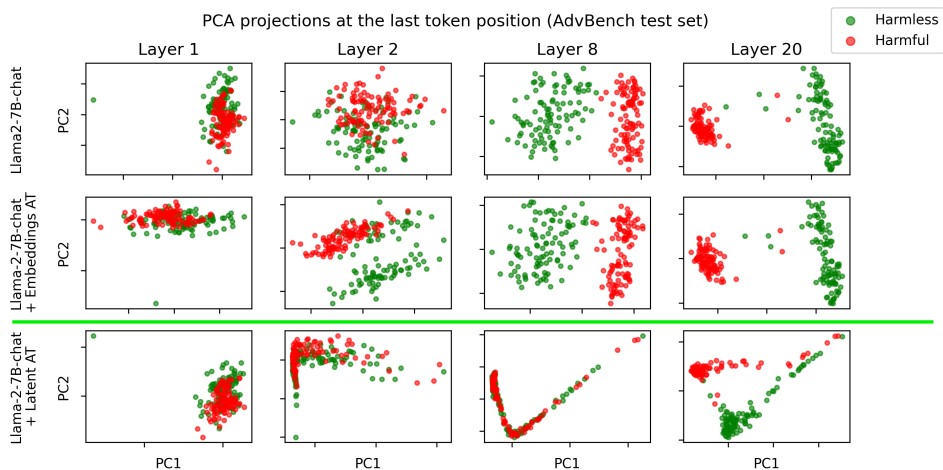

Figure 3: Principal Component Analysis (PCA) visualization of harmful vs harmless instruction representations across different network layers and model variants. Each point represents the activation pattern for a single instruction, projected onto the first two principal components. Blue points indicate harmless instructions, while red points represent harmful instructions. The plots reveal how LAT affects the separability of these instruction types in the model's latent space.

## C  LAYER-WISE ANALYSIS OF REFUSAL BEHAVIOR

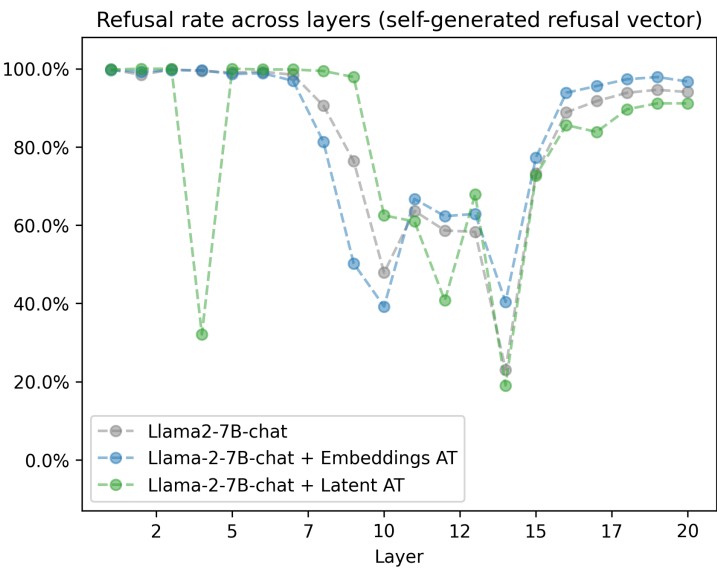

Figure 4: Layer-wise analysis of refusal rates under self-generated refusal vector attacks. The plot shows how refusal rates vary across different layers of the model architecture for the base Llama-2-7B-chat model and its Embeddings AT and LAT variants when attacked using their own refusal vectors from the same layer.

# D  STATISTICAL CONFIDENCE MEASURES OF REFUSAL RATES

Table 1: Refusal rates and statistical confidence measures across different model variants and refusal vector sources. Values in parentheses represent standard errors, and square brackets show 95% confidence intervals. All statistics are computed from a test set of 520 examples.

| Model | Refusal Vector | Refusal Rate | 95% Conf. Interval |
|---|---|---|---|
| Llama-2-7B-chat | Self-generated | 20.38% (1.77%) | [16.91%, 23.85%] |
| | From LAT | 10.77% (1.36%) | [8.10%, 13.44%] |
| Llama-2-7B-chat + Embeddings AT | Self-generated | 38.08% (2.13%) | [33.91%, 42.25%] |
| | From Baseline | 24.04% (1.87%) | [20.37%, 27.71%] |
| | From LAT | 13.65% (1.50%) | [10.71%, 16.59%] |
| Llama-2-7B-chat + LAT | Self-generated | 16.92% (1.64%) | [13.71%, 20.13%] |
| | From Baseline | 25.77% (1.91%) | [22.03%, 29.51%] |

