# OpenReview forum: "Latent Adversarial Training Improves the Representation of Refusal"
_ICLR.cc/2025/Workshop/BuildingTrust — BuildingTrust_

### Official Review · Reviewer_nAPg · 2025-02-26
**Good paper that highlights strengths / weaknesses of latent adversarial training**

**Rating:** 8
**Confidence:** 3

**Review:**

This work aims to understand how latent adversarial training (LAT) affects the representation of refusal behaviour in the LLM. The authors demonstrate that LAT improves refusal representation and provide insights into pros and cons of LAT.

Strengths:
* The paper is technically solid. The ablation attack and latent space analysis are done according to the recent sota.
* The paper discovers an important vulnerability of the LAT: “LAT’s superior encoding of refusal behavior, while potentially beneficial for model robustness, also creates a more potent attack vector.”

Weaknesses:

* No justification given to why only 14th layer is analyzed, except for the reference to Arditi et al. (2024). Would be good to discuss it, at least in the appendix.
* It would be interesting to have results on more models than llama-2, but given the nature of the paper, it’s a minor weakness.

Please fix a typo: 1.1 METHODS should be a section, not subsection.

---

### Official Review · Reviewer_4YeK · 2025-03-02
**This paper highlights Latent Adversarial Training (LAT) as a novel approach to restructuring refusal behavior in LLMs, making it more robust to external attacks but vulnerable to self-generated ones. While the study provides strong empirical insights, it lacks diverse adversarial evaluations and broader model testing, making it better suited for a Tiny Paper submission rather than a full-length**

**Rating:** 6
**Confidence:** 2

**Review:**

### Summary

This paper investigates Latent Adversarial Training (LAT) as an alternative to traditional safety fine-tuning techniques for improving LLM robustness against prompt injection and refusal manipulation attacks. LAT applies adversarial perturbations in hidden layers rather than in input embeddings, significantly altering how refusal behavior is encoded. The study finds that LAT compresses refusal behavior into fewer latent dimensions, making refusals more structured and more robust to external attacks but also more vulnerable to self-generated attacks. Evaluations using Singular Value Decomposition  and ablation attacks on LLaMA-2-7B show that LAT models transfer better across different attack vectors but fail more frequently when attacked with self-generated refusal vectors. These findings highlight both the promise and risks of LAT, suggesting future work should focus on mitigating its self-vulnerability while preserving its robustness advantages.


### Strengths

The study finds that LAT enhances robustness against external attacks but increases vulnerability to self-generated attacks, offering a nuanced perspective on its real-world applicability

The study employs Singular Value Decomposition and Principal Component Analysis to analyze how LAT restructures refusal behavior in LLMs .

It provides quantitative evidence that LAT concentrates refusal representations into fewer dimensions, making them more structured and transferable across models.

### Weaknesses

One of the key weaknesses of this paper is that its core contribution can be presented concisely without requiring a full-length submission. The primary findings are important but not extensive enough to warrant a long paper.

Many prompts in AdvBench explicitly ask for illegal or harmful content. But Modern jailbreak techniques use more indirect approaches. There is a need for more comprehensive  evaluation by using more diverse jand real-world jailbreak datasets like hack prompt dataset.

The study only evaluates LLaMA-2-7B, potentially overfitting results to a narrow model class. Additionally, the adversarial suffixes in Advbench were trained and fine-tuned using Vicuna and LLaMA-2-7B-Chat as the primary models and this can bias the results in the paper. Hence, there is a need to conduct experiments on other models like GPT, Claude, Gemini, Mistral etc.

---

### Official Review · Reviewer_7haj · 2025-03-04
**Interesting findings of how refusal representations change under LAT, but would benefit from more experimental results**

**Rating:** 5
**Confidence:** 4

**Review:**

### Summary

This work shows how Latent Adversarial Training affects the refusal representation in the residual stream of an LLM. This direction is found to be more concentrated (i.e. in the first two SVD components) in the LAT model than in the base model. This results in a refusal direction that better transfers back to the base model, but also makes the LAT model more vulnerable to refusal ablation attacks.

My main criticism is with the experimental results being limited on a single older model (Llama2-7B) which in my experience can exhibit different behaviour to more modern models.  While I think the findings and insights are nice, the limited experimental results make this borderline to me. I would recommended acceptance with more comprehensive experimental results (i.e. a conditional acceptance with the inclusion of experiments on several more modern models -- I would leave this to the discretion of the meta-reviewer/ACs). Current models are often even smaller (e.g. Llama3.2-3B-instruct, Phi3.5, etc) and computing refusal vectors is not the most expensive task, so I don’t think compute should be a major limitation.

### Strengths

- Interesting findings of how LAT impacts the refusal representation in the residual stream; the result that the LAT model results in a direction that is easier to intervene on is surprising
- Transfer results between models are also interesting
- Well written and clear, easy to follow

### Weaknesses

- Limited experimental results make it difficult to tell how robust this phenomenon is. The analysis is strictly limited to Llama2-7B; while I generally object to criticisms that simply complain about running on other models, I do think this is important here because in my experience Llama 2 can behave differently to more recent LLMs, and I believe it’s important to provide more convincing experimental results.

### Questions/Comments

- Missing citation for *Efficient Adversarial Training in LLMs with Continuous Attacks* (Xhonneux et al 2024) with regards to “embeddings AT”
- Why do you think there’s a difference between LAT and embedding adversarial training? I.e. in Figure 2, why does embedding AT seem to be comparable to the base model, while LAT results in a significant increase in explained variance? Clearly LAT is more likely to directly impact the latent representations in the residual stream, but one would expect continuous attacks in the embedding space should also impact the representations in the residual stream?

---

### Decision · Program_Chairs · 2025-03-04

Accept